# Light and Affordable Vision System for Mouth Opening–Closing Movement Deviation Assessment in Women with Mild Temporomandibular Joint Disorder

Jagoda Goślińska [1],*, Agnieszka Wareńczak-Pawlicka [1], Jarosław Gośliński [2], Piotr Owczarek [2] and Przemysław Lisiński [1]

1. Department of Rehabilitation and Physiotherapy, University of Medical Sciences, 28 Czerwca 1956 Str., No 135/147, 60-545 Poznan, Poland; a.warenczakpawlicka@ump.edu.pl (A.W.-P.); plisinski@vp.pl (P.L.)
2. Aisens Sp. z o. o., Lubeckiego 23A, 60-348 Poznan, Poland; j.goslinski@aisens.co (J.G.); p.owczarek@aisens.co (P.O.)
* Correspondence: jgoslinska@ump.edu.pl

**Abstract:** (1) Background: This paper aims to demonstrate a novel and, in terms of application, universal method of the qualitative and quantitative assessment of mandibular movement. (2) Methods: Mandibular movements are recorded by a mono-vision system where a digital camera is fixed on a special extension arm attached to the patient's head. The described method was used to check lateral deviations in 25 women with mild temporomandibular joint disorder (TMD) and in 25 women from the control group. The maximum deviation (MAX) and mean deviation (MSE) parameters were examined. In addition, the occurrence of parafunctions and joint clicking in the temporomandibular joints was checked in the examined individuals. (3) Results: Among the women with mild TMD, a significantly higher MSE parameter was found compared to the control group. Joint clicking also occurred significantly more frequently. The parameter of MAX did not differ significantly between the examined groups. (4) Conclusions: In women with mild TMD, a lack of movement coordination between the right and left joints can be observed. The method's high availability combined with simple use makes it a valuable tool for experts in different fields who diagnose and treat stomatognathic system disorders.

**Keywords:** mandibular movements; mandibular lateral deviation; optical motion capture; temporomandibular joint

## 1. Introduction

It is estimated that problems with the masticatory apparatus are experienced by about 31% of the population [1], and women are the majority [2,3]. The problem is primarily observed between the ages of 20 and 40, which corresponds to young adults [1]. Additionally, this issue is also noticed in children at a rate of 11% [1]. Furthermore, the impact of the COVID pandemic on the prevalence of temporomandibular disorders (TMD) has been observed. Most researchers suggest that the increased number of individuals with TMD may be due to isolation during the pandemic, which can contribute to depression, irritability, and limited access to specialists [4,5]. On the other hand, Haddad et al. suggest that TMD could be one of the symptoms of a COVID-19 infection [6]. However, it is difficult to definitively determine since the etiology of TMD is multifactorial. The causes of TMD may include occlusal problems (e.g., lack of balanced posterior occlusal support), muscle tension, parafunctions, emotional stress, unilateral chewing, or tooth loss [7,8]. However, it appears that multiple causes often occur simultaneously [7,9,10]. As evidenced by relevant literature [2,11], the complexity of this type of dysfunction makes people seek comprehensive medical help, which combines the efforts of stomatologists, physical therapists, and

psychologists [12]. One of the many symptoms presented by patients is qualitative and quantitative impairments of mandibular movements.

The range is the most common parameter evaluated in the context of movement. However, it should be noted that in addition to the physiological range of motion, the quality of jaw movement is also important, which is dependent on the coordination between the movement of right and left temporomandibular joints. This coordination is influenced by many factors, such as the proper functioning of the chewing muscles, tension of the joint capsule, degree of damage to the joint surfaces, and position of the intra-articular disc. Physiotherapy should be able to address most of these causes. Physiotherapy focused on TMD primarily relies on exercises and manual therapy, including the massage of the jaw and neck muscles, mobilization of the temporomandibular joints, and posture re-education [12,13].

Numerous methods for assessing mandibular motion are dedicated especially for dentists [14–18]. Objective methods require measuring tools, which exhibit different levels of accuracy, different set-up methods, and different requirements towards the room and staff qualifications. The use of these measuring tools in practice is limited by high purchase costs and application time. As a result, for the needs of physiotherapy, lateral deviations from the correct track during abduction are often assessed visually [19,20]. Vision system-based methods provide an alternative. In most cases, they are easier to use, cheaper, and more tolerable for patients [14,21].

For the purpose of the functional assessment for the needs of physiotherapy, it appears that visual systems seem to be a better solution. Physiotherapy needs a quick and accessible way to check if the quantity and quality of jaw movement have improved. After analyzing the relevant literature, it has been concluded that there exists a need for constructing a low-cost tool that would be easy to use and sensitive enough to be successfully employed by physical therapists. For this reason, the study aims to present a new device for studying mandibular movements and to use it to measure the occurrence of lateral deviations of the jaw during the movement of opening and closing the mouth.

## 2. Materials and Methods

The study involved 50 women aged 20–55, of which 25 women with temporomandibular joint dysfunction were included in the experimental group. The control group consisted of the remaining 25 women who did not exhibit any signs of TMD. All participants provided written consent for the research. The study has been approved by the Bioethical Committee of the Poznan University of Medical Sciences (Resolution No. 501/14). There were no statistically significant differences in age between the patients in the respective subgroups (*t*-test, $p = 0.823$). Table 1 presents the age characteristics of the study.

**Table 1.** Main anthropometric data of the study population (Student's *t*-test).

| Group | No. | Mean $\pm$ SD | Median | Min–Max | *p*-Value |
|---|---|---|---|---|---|
| Experimental | 25 | 40.0 $\pm$ 7.4 | 42 | 24–53 | 0.813 |
| Control | 25 | 33.9 $\pm$ 7.5 | 33 | 23–46 | |

In the preliminary qualification for the study, the clinical part of the Helkimo index was used [22]. In the study group, women were categorized as Group II or III based on the provided index, whereas women in the control group were categorized as Group I. The exclusion criteria for the study group included:

- Age below 20 or above 55 years;
- Males;
- Rheumatic, neoplastic, and central nervous system diseases;
- Lack of more than 5 teeth (excluding third molars);
- Neuralgia of cranial nerves;
- Previous injuries to the skull, jaw, or cervical spine;

-      Previous surgical treatment of the skull or jaw;
-      Joint disc blockage without reduction;
-      Angle's Class III malocclusion;
-      Open bite.

The study consisted of a clinical examination and a camera examination.

### 2.1. Clinical Examination

During the clinical examination, the presence of joint clicking during the opening and closing of the jaw without load on the right and left sides was evaluated. The examination involved placing first the index finger and then the stethoscope in front of the ear at the level of the temporomandibular joint, and checking for the occurrence of clicks in the joint. Additionally, patients were asked about the occurrence of oral parafunctions.

### 2.2. Camera Recording

To examine the trajectory of the mandible during opening and closing, vision analysis was employed and performed PROWEBCAM C920 (Logitech, Lausanne, Switzerland) digital camera that featured 1920 × 1080 resolution (at 30 fps) and was fixed on an aluminum extension arm attached to a headset worn by the subject (Figure 1).

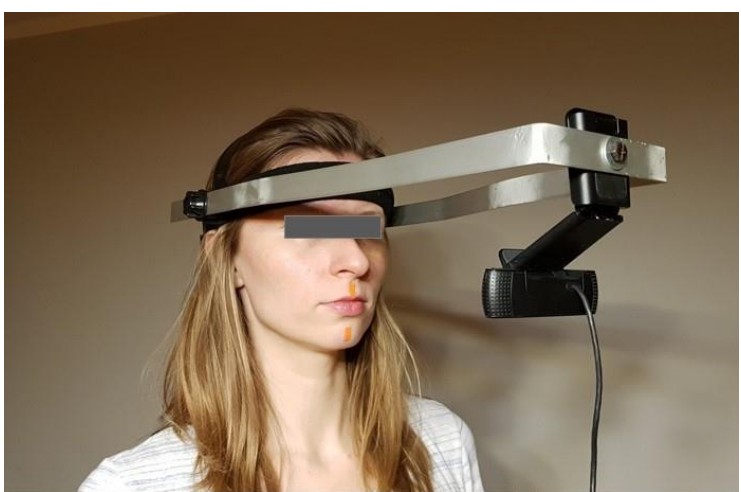

**Figure 1.** A digital camera connected to an adjustable headgear through an aluminum extension arm.

The camera was aligned parallel to the subject's face at a distance of 0.3 m. The device featured a high-resolution charged-coupled device matrix. Product software allowed for automatic sharpness corrections of image foreground. The study used two 10 mm × 5 mm reference markers, which were accurately reproduced on the camera-recorded images. The markers were pieces of tape used in kinesiology taping therapy; the material is FDA-certified and registered as a medical product. The markers were orange, and the color differentiated them from the surrounding image background (the face, walls, and proximity). As a result, it was possible to correctly detect the markers' real geometric center and track the mandible's movement trajectory based on captured and calculated marker positions. In line with the methodology applied by Adly et al. [14], one marker was put on the medial cleft above the upper lip, and the other one was positioned below the first one in a straight line in the middle of the chin.

Before recording, a line was displayed in the middle of the screen. This line was a reference for the correct positioning of the headset and markers. The patient was asked to open and close his mouth several times as much as possible. These initial repetitions were not recorded. Then, the recording was turned on, and the patient was asked to repeat the movement twice more. The program in the computer calculated the range of maximum

deviation from the reference (middle) line (MAX) in the frontal plane (ROD), and the mean deviation from the reference line raised to a square per one second of the recording (MSE).

Stages of Image Analysis: Software

Marker characteristics were extracted using OpenCV 4.5.0 software and an industrial vision system called Adaptive Vision Studio. The software enabled the processing of camera images and files. All result data were displayed in real-time. To use the algorithm's maximum capacity, only basic image filters were applied, such as image smoothing with Gaussian filtering, hue saturation value (HSV) image thresholding, splitting image regions into so-called BLOBs (Binary Large Objects), and finally computing the markers' mass center (Figure 2).

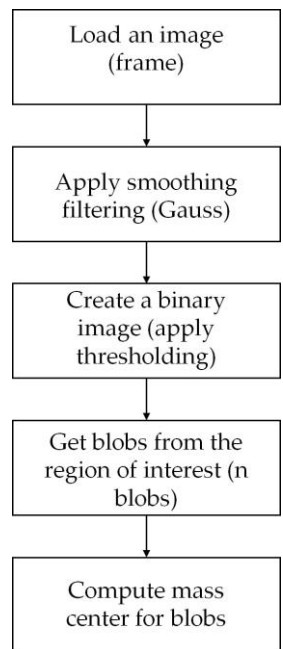

**Figure 2.** The applied image-filtering process.

Manipulation of the HSV color space (i.e., hue, saturation, and value) enabled controlling set points for filtering out image background (about RGB image). The three values' ranges were as follows: H: 0.360, S: 0.255, V:0.255. In the example below, the setpoints were H: 4.27, S: 170.255, V: 58.149. This completely removed the image background and the output showed only the identified markers (Figure 3).

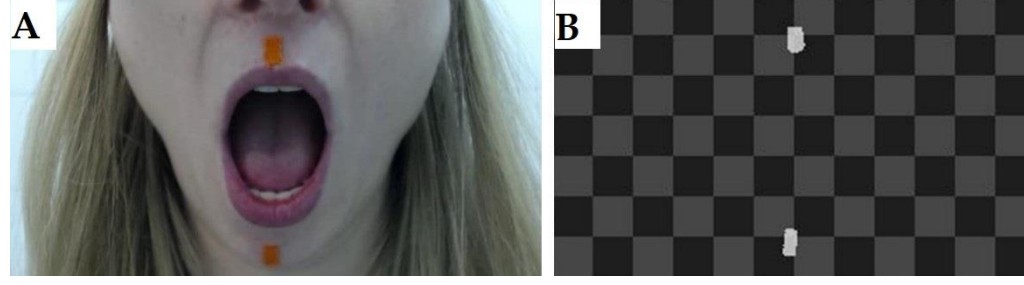

**Figure 3.** (**A**) Input the image before filtering; (**B**) Output after filtering out the image background illustrates the final outcome of using the algorithm: properly detected markers and their center points.

Figure 4 illustrates the final outcome of using the algorithm: properly detected markers and their center points.

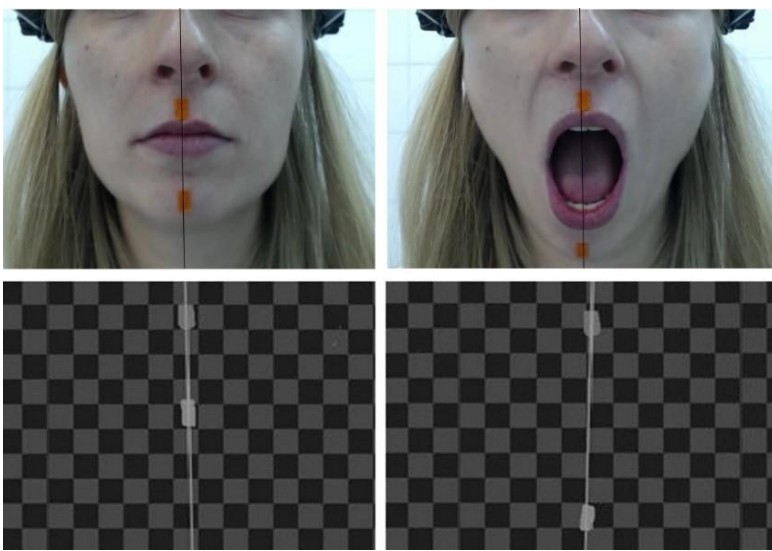

**Figure 4.** Correctly detected markers.

During measurement, data were saved in the formats of x, y, dx, and dy for each marker in each frame of the processed image, where X represents the position of a given marker's mass center on the *X*-axis, Y is the position of a marker mass center on the *Y*-axis, whereas dx and dy represent marker size on the *X*- and *Y*-axis, respectively. The referential coordinate system was placed in the upper left corner of the image as processed in OpenCV. It was a right-handed coordinate system, and its axes were oriented the following way: *X*-axis was right-oriented, *Y*-axis was downward, and *Z*-axis was inward-oriented. The x, y, dx, and dy parameters were recorded in pixels. A parameter for scaling point coordinates of the image was expressed in mm/pixel and was derived from the following relation:

$$s_c = \frac{dx_r}{\sum_{i=1}^{n} \frac{dx_i}{n}} \tag{1}$$

where $dx_r$ denoted the real value of marker size on *X*-axis, i.e., 5 mm; $dx_i$ was the measurement of the marker's size on the *X*-axis and was expressed in pixels; $n$ was several measurements taken into consideration when computing the scaling value; and $s_c$ value was determined only using the upper marker.

To establish the mandible's trajectory, f(x) line was drawn between the positions of the two markers to connect the two points. The line's parameters were calculated only during initial measurements and were constant throughout the study. With the line joining the two markers, i.e., two mass centers, it was possible to correct the image for rotations of a face in relation to the camera. Importantly, the camera did not change its position or orientation during the study. Therefore, the width of the mandible (the distance to the camera) was changing when the mouth was opening or closing. Due to the different distances of the upper and lower marker on the *Z*-axis, it was assumed that the upper one would serve as a reference; the lower marker was, in this case, a projection from its plane tangent to the upper marker's plane. As the distance of the lower marker changed (on the *Z*-axis—width), its size decreased; however, its mass center did not change position, so the same scaling parameter was used for the upper marker.

The determined parameter value of the straight line connecting markers' mass centers was used to establish an optimal movement trajectory of the mandible (assuming that when discomfort occurred during movement, the mandible remained in the same position on the *X*-axis).

Analysis of movement included an algorithm for detecting the earliest stage of opening and closing the jaw. Next, movement parameters were computed together with differ-

ences between the real and referential trajectory. An example movement pathway with highlighted start and endpoints of the mouth opening and closing is illustrated in Figure 5.

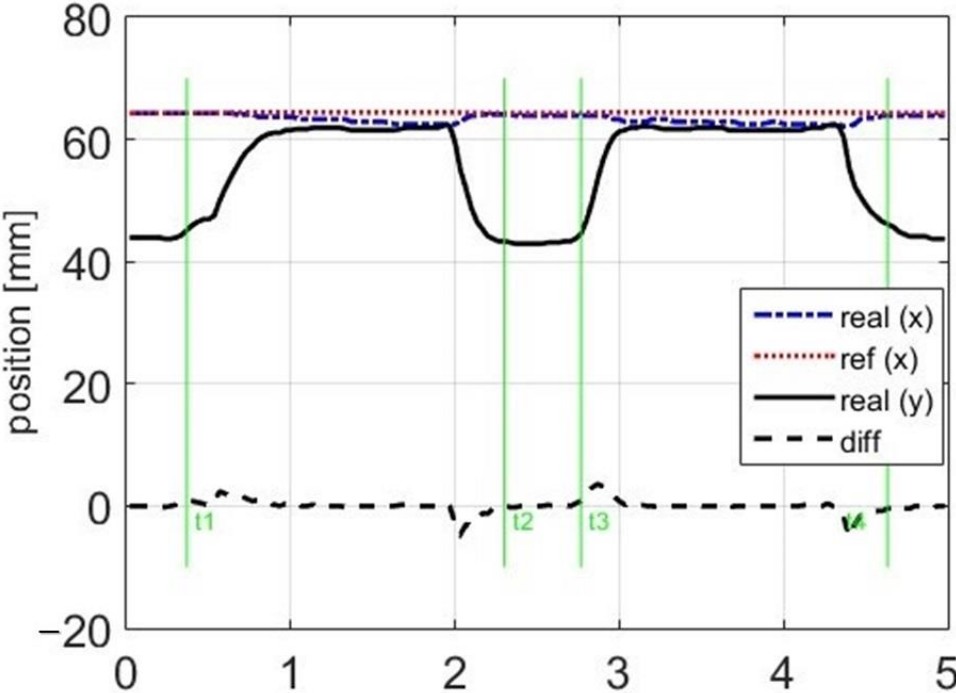

**Figure 5.** An example movement pathway with highlighted start and end points of the mouth opening. The real (x) line represents lateral displacements from the optimal path, i.e., a referential line (ref (x)). Line y represents a range of the mouth opening. Information about the f(x) line and the real value of y (real (y)) were used to determine the referential value on *X*-axis (ref (x)), which was then compared with the real value on *X*-axis (real (x)). To identify the characteristic points (t1, t2, t3, t4) of pathways for two cycles of opening and closing the mandible, a derivative of the y signal (black dashed line (diff)) and an instantaneous y value were used.

Information about the f(x) line and the real value of y (black line) was used to determine the referential value on *X*-axis (red line), which was then compared with the real value on *X*-axis (blue line).

To identify the characteristic points of pathways for two cycles of mouth opening and closing, a derivative of the y signal (black dashed line) and an instantaneous y value were used. The characteristic points were described as t1, t2, t3, and t4. Movement statistics were calculated only for signals within t = t2 − t1 and t = t4 − t3. Initial filtering used median filters, and each estimation provided the following parameters: maximum jaw abduction, normalized mean squared error in the frontal plane, and maximum displacement in the frontal plane. Only the value of maximum opening and normalized squared error was determined during t = t2 − t1 + t4 − t3. The most significant parameter was the value of normalized mean squared error, which was calculated by dividing the sum of squared errors in movement trajectory on the *X*-axis by the number of measurements taken during analysis time t. The quality of the image and vision signal influenced estimation error, which depended on many factors, such as exposure and selection of filter parameters when extracting marker characteristics off the image.

## 3. Results
### 3.1. Statistics

The results were analyzed using the statistical software package STATISTICA version 13.1. In descriptive statistics, for each quantitative variable, the mean, standard deviation (SD), minimum (min), maximum (max), and median were calculated while parameters

expressed on a nominal scale were described by the number of patients and percentages in each category. The Shapiro–Wilk test was used to assess the normality of the parameters' distribution. When normality was confirmed in the case of homogeneous variances, the Student's $t$-test was used. The homogeneity of variance was tested using Levene's test. The non-parametric Mann–Whitney test was used when normality was not confirmed. The relationship between individual parameters was examined using the Pearson linear correlation coefficient (r). All statistical hypotheses were verified at a significance level of $p < 0.05$. The dependence of parameters expressed on a nominal scale was tested using the chi-squared test.

### 3.2. Clicking in Temporomandibular Joints

During the clinical examination, the presence of clicking in the temporomandibular joints on the right and left sides was evaluated. The studies showed that clicking occurred significantly more often on both sides in the experimental group (Table 2). Eleven individuals exhibited clicking in one of the joints, and bilateral clicking was found in two women.

**Table 2.** Comparison of clicking in right and left temporomandibular joints between the experimental and control group (chi$^2$ test).

| Opening with Clicking | Answer | Experimental N = 25 | Control N = 25 | $p$-Value |
|---|---|---|---|---|
| Right side | yes | 6 | 0 | 0.03 |
| | no | 19 | 25 | 0.4 |
| Left side | yes | 9 | 0 | 0.004 |
| | no | 16 | 25 | 0.4 |

The $p$-value when comparing the experimental group to the control group.

### 3.3. Oral Parafunctions

In all women in the study group, at least two parafunctions were present while all women in the control group did not report any consciously performed abnormal movement habits of the masticatory system. The frequency of occlusal parafunctions, including grinding and clenching, as well as non-occlusal asymmetrical parafunctions in the studied groups is presented in Table 3.

**Table 3.** Comparison of the parafunctions in the experimental and control group.

| Parafunction Type | Answer | Experimental N = 25 | Control N = 25 |
|---|---|---|---|
| teeth clenching | yes | 24 | 0 |
| | no | 1 | 25 |
| teeth grinding | yes | 12 | 0 |
| | no | 13 | 25 |
| Asymmetrical non-occlusal | yes | 25 | 0 |
| | no | 0 | 25 |

It was found that at least one asymmetrical non-clenching parafunction was present in the study group. However, no significant correlation was found between the number of non-clenching parafunctions and MSE ($p > 0.05$).

### 3.4. Mandibular Deviation Parameters

The study found no differences between the experimental and control group in the maximum deviation of the jaw from the standard curve during the opening and closing of the mouth. However, significant differences were observed in the MSE parameter. Specific details are presented in Table 4.

**Table 4.** Comparison of the range of two mandibular deviation parameters—MAX and MSE between experimental and control groups.

| Parameter | Group | No. | Mean $\pm$ SD | Median | Min–Max | *p*-Value |
|---|---|---|---|---|---|---|
| MAX | Experimental | 25 | 2.08 $\pm$ 0.80 | 2.01 | 0.80–4.01 | 0.113 * |
| | Control | 25 | 1.74 $\pm$ 0.67 | 1.69 | 0.62–3.17 | |
| MSE | Experimental | 25 | 52.89 $\pm$ 55.77 | 26.84 | 4.49–254.58 | 0.007 ** |
| | Control | 25 | 28.16 $\pm$ 36.01 | 12.37 | 1.67–148.11 | |

*p*-value: the comparison of intergroup age, *—Student's *t*-test, **—Mann–Whitney test.

## 4. Discussion

The measurement method outlined in this paper uses a vision system to record the movements of two markers fixed on a patient's face. The movement quality during jaw opening was assessed by registering deviations from the standard curve in the frontal plane in individuals with mild TMD compared to healthy individuals.

In the study, two parameters were assessed—the maximum deviation occurring during the entire opening and closing cycle and the normative deviation. The results indicated that the maximum deviation from the curve was not significantly greater than in the control group. Significant deviations in jaw movements may be caused by a displaced articular disc, which, if repositioned, will cause the jaw to return to its proper path, but if not, it may be the cause of a constant displacement of the jaw during movement towards the joint with a locked disc [23–27]. The subjects in the study exhibited significantly more clicking in the right and left joints than the control group. However, since our study did not register significant differences in maximum deviation, this may be due to a lack of advanced changes in the temporomandibular joint disc, which quickly reduces despite hypermobility or slipping from the head of the jaw and does not result in large lateral deviations.

In the study, we observed that the mean deviation was significantly different in individuals with mild TMD compared to healthy individuals. The mean deviation is a parameter that, unlike the maximum deviation, does not indicate the maximum lateral deviation of the jaw during opening, but it qualitatively evaluates the movement of abduction and adduction. The results of this parameter show us how much the jaw does not maintain its correct path but deviates laterally to one side or the other. Therefore, since there were no differences in the maximum range of deviation, but there was a significant difference in mean deviation between the study and control groups, there were more than one deviation from the standard curve, but each one was quite small.

It seems that the cause of differences in mean deflections was not likely due to problems with the disc, which, although it was in dislocation in 60% of patients, quickly reduced. Differences in normative deflections could also be suspected in people who have bilateral disc problems, but among the examined patients, there were only two such cases.

Bae and Park suggest that this may be due to a lack of muscular coordination between the right and left sides [28]. We suspect that this may be related to the occurrence of parafunctional activities of the masticatory system. Parafunctions, as non-physiological activities habitually performed within the masticatory system, usually occur against the background of stress [29] and can be asymmetric [30,31]. Examples of parafunctional activities that can be included here are: sucking or biting the lip or cheek on one side, biting objects, or playing with the tongue [30,32]. It would seem, therefore, that habitually repeated asymmetric activities could affect deviations in jaw movement. In our study, all individuals in the study group had at least one asymmetric parafunctional activity of the masticatory system. Nevertheless, we did not observe a correlation between the number of asymmetric parafunctions and normative deviation. Interestingly, Baba et al. [33] suggest that one-sided clenching caused by uneven contacts on the right and left sides causes greater tension in the temporal muscle on that side and changes in jaw movement. We did not find studies by other authors that indicate a relationship between lateral deviations of the jaw and the occurrence of parafunctions.

The aim of our study was also to present a new method for assessing mandibular movement. We assume that the method may constitute an attractive alternative to other methods for measuring the mouth opening–closing movement, especially useful for a physiotherapist. The proposed method may be compared with existing ones by focusing on a few aspects.

First of all, the way measuring equipment is attached to a patient's head seems relevant for both the patient's comfort and obtained results [15,17,18]. The method we are proposing uses a camera connected to adjustable headgear through an aluminum extension arm. Besides markers placed on the medial cleft and chin, no other element of the equipment was in contact with the area measured. Additionally, the patient did not report any discomfort related to the equipment's weight or attachment.

The material markers were made of a proven good choice: it was safe for the patient, was easily and firmly fixed to the skin, and, most importantly, had a distinct color that the camera could simply isolate from the background. Other researchers used pieces of metal or simple paper cubes to act as markers [14,15]. We placed the markers in a straight line. The first one was on the medial cleft, and the second one was in the middle of the chin. Adly et al. [14] used the same marker positions. In many other methods, the main marker had direct contact with the inferior dental arch, which was supposed to eliminate the influence of skin movement during opening and closing to raise accuracy [15,16,34]. To achieve such positioning, plastic splints [35,36,36,37] or thermoplastic material [14,15] was applied. The authors of these methods admit that such marker attachment results in improved measurement accuracy; nevertheless, it potentially affects the proprioceptive system and the patient's comfort, which may also have an impact on results [15,18].

Available methods examine the mouth opening–closing trajectory by assessing the maximum range in two or three planes, depending on the method [37,38]. Besides the MAX, our method also includes an MSE parameter. It is an analysis of the entire cycle from initiating opening to closing the mouth; the parameter informs us about the degree the mandible deviated from the optimal trajectory per one second. The disadvantage of our method is the evaluation of movement only in 2D, so it is not possible to differentiate the range of rotation and translation when opening the mouth. However, this method gives the possibility to evaluate the coordination between the right and left temporomandibular joint through the maximum deviation from the reference line and MSE parameter, which is mean square error from the ideal reference track. The need for assessing the deviation of the mouth opening and closing was also noted by other researchers [21,39].

Another problem that needs to be addressed in vision systems used for recording movements of the mandible is measurement accuracy. The relevant literature contains many methods proven to deliver very high accuracy. A paper by Furtado et al. [15] compares the accuracy levels of different equipment, and in almost all cases, they are in the region of tenths of a millimeter. Our method does not produce such accuracy, as the measurement error is estimated at less than 1 mm (the error was assessed by computing the error of movement of the upper marker when the conditions were steady, i.e., when the position of the marker in the camera coordinate system of reference was not changing). This may be considered a drawback, although the priority for us in developing this method was the easy and fast use as well as low invasiveness. However, we are aware that for orthodontic or prosthetic purposes, such accuracy is insufficient. Nonetheless, we believe that the proposed method may be used in different treatment stages, e.g., to evaluate physical therapies improving the range of motion and trajectory of mouth opening. After all, our method seems to also be much more objective than a visual examination.

Another advantage of our method is that it may be applied in almost any condition. The measuring equipment is light and takes up little space. Therefore, an examination may be conducted in almost any place. There only has to be good lighting in the room. Other methods often use a few cameras, which need to be installed in relation to each other in specific positions [14,18,21,35,36]. Additionally, other methods often require the use of head-stabilizing elements [15,35,38]. These characteristics will often translate to specific

requirements for the room, and the equipment itself will have to be permanently stored where examinations are performed.

The most advanced methods available on the market, such as WinJaw or MODJAW, comprehensively assess jaw movements and are characterized by high accuracy and a three-dimensional motion analysis [40–42]. This is a clear advantage compared to the method proposed by us. However, these mentioned methods are justified for the work of a dentist, which requires great precision in planning orthodontic or prosthetic treatment. From the perspective of a physiotherapist, it is necessary to use a method that minimally interferes with muscle function and provides a quick preliminary diagnosis of the jaw movement pattern, allowing for subsequent evaluation of the effects of physiotherapy. In this regard, our proposed method appears to be sufficient, and due to its low cost, it may become a popular tool for assessing jaw movement.

*Limitation*

The presented study also has some limitations. It would be valuable to investigate the protrusion and lateral movements of the jaw, which would also be beneficial for physiotherapy. Furthermore, it would be beneficial to use a different scale during patient recruitment, such as the RDC/TMD [23], which is more detailed. Then, the research group could be even more homogeneous.

## 5. Conclusions

In summary, based on our study, several conclusions can be formulated:

1. In women with mild TMD, a lack of coordination between the right and left joints can be observed;
2. In women with mild TMD, a displaced disc is reduced in the early stage of mouth opening, so it does not affect the range of maximum lateral deviations;
3. The proposed vision system method for assessing the quality of mandibular opening and closing movements appears to be an interesting tool for evaluating the coordination between the left and right temporomandibular joints;
4. With easy and fast attachment, the proposed method lends itself to population-based diagnostic studies and may be successfully used even by individuals with no advanced knowledge of the masticatory apparatus;
5. The proposed device may have applications in physiotherapy; however, it is not sufficient for use in dental or surgical treatments.

**Author Contributions:** J.G. (Jagoda Goślińska) and J.G. (Jarosław Gośliński) designed the testing method. J.G. (Jagoda Goślińska) recruited participants, carried out a clinical assessment, interpreted the data and prepared the first version of the manuscript. J.G. (Jarosław Gośliński) and P.O. proceeded with all technical aspects of the study. A.W.-P. was responsible for statistics. P.L. reviewed and contributed substantive inputs to the final version of the manuscript. All authors have read and agreed to the published version of the manuscript.

**Funding:** This research received no external funding.

**Institutional Review Board Statement:** The study was conducted in accordance with the Declaration of Helsinki and approved by the Bioethical Committee of the Poznan University of Medical Sciences (Resolution No. 501/14). The clinical trial has been registered in a public repository (number: NCT03571035).

**Informed Consent Statement:** Informed consent was obtained from all subjects involved in the study.

**Data Availability Statement:** Data is unavailable due to privacy.

**Conflicts of Interest:** The authors declare no conflict of interest.

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
