# Peer review of "Light and Affordable Vision System for Mouth Opening–Closing Movement Deviation Assessment in Women with Mild Temporomandibular Joint Disorder"

_applsci, doi:10.3390/app13148224_

Round 1

Reviewer 1 Report

Dear Authors, 

thank you for the paper I had an opportunity to review. Here are some suggestions of mine:

1. In the 1st paragraph of introduction, please add the information on the geographic and demographic frequency of TMJ problems and if the Covid pandemic influenced the occurence of TMD in population.

- eg. Emodi-Perlman A, Eli I. One year into the COVID-19 pandemic – temporomandibular disorders and bruxism: What we have learned and what we can do to improve our manner of treatment. Dent Med Probl. 2021;58(2):215–218. doi:10.17219/dmp/132896

- Add some general information on TMD causes, eg. 

Boening K, Wieckiewicz M, Paradowska-Stolarz A, Wiland P, Shiau YY. Temporomandibular disorders and oral parafunctions: mechanism, diagnostics, and therapy. Biomed Res Int. 2015;2015:354759. doi: 10.1155/2015/354759. 

- Did you refer to mental health, eg.

FlorjaÅ„ski W, Orzeszek S. Role of mental state in temporomandibular disorders: A review of the literature. Dent Med Probl. 2021;58(1):127–133. doi:10.17219/dmp/132978

Please, find more references on that topics

2. Please, redraft the introduction section so that all the parts combine more. I would add what exactly do you mean by physiotherapy (see: massage, biofeedback, kinesiotaping etc).

3. One of the first sentences in M&M section should be the ethical commitee approval number

4. Line 79 - please, modify this sentence - it is not too fortunate

5. line 83 - which teeth do you mean? Were 3rd molars included into a total number? does this criterion include teeth removed from orhodontic reasons?

6. Line 103 - the company should be in brackets, the country of producion should be added

7. the figure 2 is totally unclear. Is it neccesary? If so, this should contain more details on how the study (image filtirng) was conducted.

8. More details on the imaging (incl. format etc) should be added

9.  Were the TMJ symptoms recorded only when free joint movement was applied? Or were the TMJs loaded somehow? If so, which tests did the authors use

10. In the discussion, refer to:

- ModJaw system

- use of articulators

11. remove the dod in line 262 - before the cited references

12. Add the limitations of the study.

Thank you

Author Response

Thank you for your feedback. We appreciate your comments. We highlighted in red colour all the changes we have made in the manuscript. Please find our responses below, where we explain things that we corrected.

Comment 1. In the 1st paragraph of introduction, please add the information on the geographic and demographic frequency of TMJ problems and if the Covid pandemic influenced the occurence of TMD in population.

- eg. Emodi-Perlman A, Eli I. One year into the COVID-19 pandemic – temporomandibular disorders and bruxism: What we have learned and what we can do to improve our manner of treatment. Dent Med Probl. 2021;58(2):215–218. doi:10.17219/dmp/132896

- Add some general information on TMD causes, eg. 

Boening K, Wieckiewicz M, Paradowska-Stolarz A, Wiland P, Shiau YY. Temporomandibular disorders and oral parafunctions: mechanism, diagnostics, and therapy. Biomed Res Int. 2015;2015:354759. doi: 10.1155/2015/354759. 

- Did you refer to mental health, eg.

FlorjaÅ„ski W, Orzeszek S. Role of mental state in temporomandibular disorders: A review of the literature. Dent Med Probl. 2021;58(1):127–133. doi:10.17219/dmp/132978

Please, find more references on that topics

Respond: We changed the introduction according to your guidelines. We provided a more detailed description of epidemiological data in the context of TMD, taking into account the COVID-19 pandemic. Additionally, we included information regarding the multifactorial etiology of TMD.We have also utilized additional references, including those suggested by you.

Comment 2. Please, redraft the introduction section so that all the parts combine more. I would add what exactly do you mean by physiotherapy (see: massage, biofeedback, kinesiotaping etc).

Respond: We added information about physiotherapy in TMD.

Comment 3. One of the first sentences in M&M section should be the ethical commitee approval numer

Respond: We relocated the information about the ethics committee based on your suggestion.

Comment 4. Line 79 - please, modify this sentence - it is not too fortunate

Respond: Thank you for that note. We have changed it.

Comment 5. line 83 - which teeth do you mean? Were 3rd molars included into a total number? does this criterion include teeth removed from orhodontic reasons?

Respond: Thank you very much for the feedback. We indeed overlooked that clarification. We have now added a comment to address it.

Comment 6. Line 103 - the company should be in brackets, the country of producion should be added

Respond: We have added details that you mentioned.

Comment 7. the figure 2 is totally unclear. Is it neccesary? If so, this should contain more details on how the study (image filtirng) was conducted.

Respond: We have changed Figure 2, and we hope that it is now more clear.

Comment 8. More details on the imaging (incl. format etc) should be added

Respond: We added th number of frames per second.

Comment 9.  Were the TMJ symptoms recorded only when free joint movement was applied? Or were the TMJs loaded somehow? If so, which tests did the authors use

Respond: We only examined clicks during jaw opening without any load. We have added information about it.

Comment 10. In the discussion, refer to:

- ModJaw system

- use of articulators

Respond: Thank you for the suggestion. We referred to the modJaw system. However, we did not specifically address articulators as they are predominantly used in the context of tooth contacts during various jaw movements. Our device does not involve the positioning of teeth. Do you think it would still be worthwhile to mention these differences?

Comment 11. remove the dod in line 262 - before the cited references

Respond: Thank you for your alertness. We have changed it.

Comment 12. Add the limitations of the study.

Respond: We added chapter about limitation.

Reviewer 2 Report

I suggest you move the statistics part from the material and method chapter to the result chapter. 

The article focuses on a new technique used in the joint examination, so I consider that conclusion number 1 and number 2 are not relevant to this study.

Author Response

Thank you for your feedback. We appreciate your comments. We highlighted in red colour all the changes we have made in the manuscript. Please find our responses below, where we explain things that we corrected.

Comment 1. I suggest you move the statistics part from the material and method chapter to the result chapter.

Respond: Thank you for that note. We have changed it.

Comment 2. The article focuses on a new technique used in the joint examination, so I consider that conclusion number 1 and number 2 are not relevant to this study.

Respond: It is important for us to include the first two conclusions since they provide valuable clinical implications from the conducted study.

Reviewer 3 Report

Thank you for this manuscript.  I appreciate the attempt to try analyze a very complex problem like TMDs. 

I think it can be interesting, but i really suggest to highlight in the manuscript, in the introduction, discussion and conclusion the RDC to transmit the idea, which the literature already says, that this is a multifactorial issue.

I do not think TMDs are just connected with the oral system and OCCLUSION. This can be a factor, but not the only one. So i would say that the patients before being included in this study should have been analyzed better through DC.

To Be clear, i would like if you improve introduction, discussion and conclusion, highlighting the multifactorial etiology of TMDs.

The English is good, i would just check some sentence.

Author Response

Thank you for your feedback. We appreciate your comments. We highlighted in red colour all the changes we have made in the manuscript. Please find our responses below, where we explain things that we corrected.

Comment 1. I think it can be interesting, but I really suggest to highlight in the manuscript, in the introduction, discussion and conclusion the RDC to transmit the idea, which the literature already says, that this is a multifactorial issue.

 Respond: In the introduction we add about the multifactorial etiology of TMD.

Comment 2. I do not think TMDs are just connected with the oral system and OCCLUSION. This can be a factor, but not the only one. So i would say that the patients before being included in this study should have been analyzed better through DC.

Respond: Indeed, it would be valuable to consider the most common differential diagnoses of RDC during recruitment. However, we decided to use the Helkimo Index due to the relatively early stage of dysfunction observed in our patients.

Comment 3. To be clear, I would like if you improve introduction, discussion and conclusion, highlighting the multifactorial etiology of TMDs.

Respond: We tried to changed it according to your guidelines.

Round 2

Reviewer 1 Report

Thank you for the corrections and responsw. In this form, the article could be accepted

Author Response

Dear Reviewer, 

Thank you for the positive evaluation and for conducting the review.

Reviewer 3 Report

I appreciate you tried to improve the manuscript, but you did deepen the the occlusal and bio psycho social theory. 

You need to check English. For example line 73. I would add this two references: one about TMD and covid and another  one about physiotherapy and TMD. 

Saccomanno S, Saran S, De Luca M, Mastrapasqua RF, Raffaelli L, Levrini L. The Influence of SARS-CoV-2 Pandemic on TMJ Disorders, OSAS and BMI. Int J Environ Res Public Health. 2022 Jun 10;19(12):7154. doi: 10.3390/ijerph19127154. PMID: 35742398; PMCID: PMC9222869.

Saran S, Saccomanno S, Petricca MT, Carganico A, Bocchieri S, Mastrapasqua RF, Caramaschi E, Levrini L. Physiotherapists and Osteopaths' Attitudes: Training in Management of Temporomandibular Disorders. Dent J (Basel). 2022 Nov 4;10(11):210. doi: 10.3390/dj10110210. PMID: 36354655; PMCID: PMC9689146.

It needs some improvement

Author Response

Thank you for your feedback. We appreciate your comments. We highlighted in red colour all the changes we have made in the manuscript. Please find our responses below, where we explain things that we corrected.

Comment 1. You need to check English. For example line 73.

Respond: We asked the native speaker to revise manuscript.

Comment 2. I would add this two references: one about TMD and covid and another  one about physiotherapy and TMD. 

Saccomanno S, Saran S, De Luca M, Mastrapasqua RF, Raffaelli L, Levrini L. The Influence of SARS-CoV-2 Pandemic on TMJ Disorders, OSAS and BMI. Int J Environ Res Public Health. 2022 Jun 10;19(12):7154. doi: 10.3390/ijerph19127154. PMID: 35742398; PMCID: PMC9222869.

Saran S, Saccomanno S, Petricca MT, Carganico A, Bocchieri S, Mastrapasqua RF, Caramaschi E, Levrini L. Physiotherapists and Osteopaths' Attitudes: Training in Management of Temporomandibular Disorders. Dent J (Basel). 2022 Nov 4;10(11):210. doi: 10.3390/dj10110210. PMID: 36354655; PMCID: PMC9689146.

Respond: Thank You for this tip. We add these references.
